# Storage-Dependent Generation of Potent Anti-ZIKV Activity in Human Breast Milk

**DOI:** 10.3390/v11070591

**Published:** 2019-06-28

**Authors:** Carina Conzelmann, Min Zou, Rüdiger Groß, Mirja Harms, Annika Röcker, Christian U. Riedel, Jan Münch, Janis A. Müller

**Affiliations:** 1Institute of Molecular Virology, Ulm University Medical Center, 89081 Ulm, Germany; 2Guangzhou Key Laboratory of Drug Research for Emerging Virus Prevention and Treatment, School of Pharmaceutical Sciences, Southern Medical University, Guangzhou 510515, China; 3Institute of Microbiology and Biotechnology, Ulm University, 89081 Ulm, Germany; 4Core Facility Functional Peptidomics, Ulm University Medical Center, 89081 Ulm, Germany

**Keywords:** zika virus, breast milk, transmission, breastfeeding

## Abstract

Zika virus (ZIKV) causes congenital neurologic birth defects, notably microcephaly, and has been associated with other serious complications in adults. The virus has been detected in human breast milk and possible transmissions via breastfeeding have been reported. Breast milk is rich in nutrients and bio-active substances that might directly affect viral infectivity. Thus, here, we analyzed the effect of human breast milk on ZIKV infection. We observed that fresh human breast milk had no effect on ZIKV, but found that upon storage, milk effectively suppressed infection. The antiviral activity is present in the fat-containing cream fraction of milk and results in the destruction of the structural integrity of viral particles, thereby abrogating infectivity. The release of the factor is time dependent but varies with donors and incubation temperatures. The viral titer of milk that was spiked with ZIKV decreased considerably upon storage at 37 °C for 8 h, was lost entirely after 2 days of 4 °C storage, but was not affected at −20 °C. This suggests that cold storage of milk inactivates ZIKV and that the antiviral factor in milk may also be generated upon breastfeeding and limit this transmission route of ZIKV.

## 1. Introduction

Zika virus (ZIKV) is a (re-)emerging virus that was originally identified in 1947 in Uganda [1] but has now caused a series of epidemics in Micronesia [2], the South Pacific [3], and the Americas [4]. To date, 84 countries or territories reported ZIKV transmissions [5]. Initially considered a harmless infection in humans, ZIKV may cause disease in adults, including meningoencephalitis [6], myelitis [7], thrombocytopenia [8], and Guillain–Barré syndrome [9,10]. It is now established that ZIKV can be transmitted from mother to child during pregnancy, where it can cause fetal demise, microcephaly, and other congenital problems [11], which may develop in up to 46% of the cases [12]. Thus, ZIKV infection poses a high risk for human health, as vaccines and therapeutics are not available.

ZIKV is mainly transmitted to people through the bite of an infected *Aedes* species mosquito [13]. This vector-dependent route of transmission mainly restricts the ZIKV pandemic to regions where the *Ae. aegypti* and *Ae. albopictus* mosquitoes are endemic [13]. ZIKV transmission has also been reported to occur through sexual contacts [14,15,16,17], laboratory exposure and blood transfusion [15,18], or from mother to child intrauterine [15,19], intrapartum [15,20] or possibly via breastfeeding [15,20,21,22,23,24,25]. There are three reported cases of probable ZIKV transmission via breast milk [20,22,23,25], but final evidence and risk of transmission remain inconclusive [15,21,24,25]. As evidence is sparse and the health benefits of breastfeeding outweigh the transmission risk, the WHO recommends mothers with suspected, probable or confirmed ZIKV infection or in areas of ongoing ZIKV transmission to routinely continue breastfeeding [26]. Nevertheless, there are clear data that ZIKV is present in various body fluids [15,27] including breast milk, that contains ZIKV genomic RNA [15,20,21,22,23,24,25,28,29] and infectious particles [15,21,22,23,24,25,28,29]. However, although the virus can be detected and quantified by RT-PCR in breast milk and other body fluids, viral genome copy numbers do not always correlate with the infectious titer of the virus [18,20].

Breast milk is a body fluid that nurtures and protects the infant. It is rich in nutrients and vitamins and provides the child with carbohydrates, proteins, fat, minerals, hormones, growth factors and antibodies [30,31]. Breast milk can be a source of viral infection [32], but also contains bioactive substances that may directly affect viral infectivity [31,33]. A recent study analyzed the stability of ZIKV in breast milk at 4 °C and found that ZIKV is inactivated upon prolonged storage [34]. Here, we aimed to expand this finding and explored ZIKV stability at physiological temperatures and how breast milk may directly affect ZIKV infection. We show that fresh human milk had no significant effect on ZIKV infection, however, storage of milk resulted in the generation of a potent anti-ZIKV factor. Similar to earlier findings for hepatitis C virus (HCV), this factor is dominant in the fat-containing cream fraction and possibly released by lipases present in milk or gastric juice [35]. This factor rapidly abrogates infectivity by physical destruction of the viral particle and may play a role in virus inactivation upon storage of milk for later use or in the gastrointestinal tract of babies upon breastfeeding. This might explain why ZIKV transmission via breastfeeding is hardly observed.

## 2. Materials and Methods

### 2.1. Cell Culture

Vero E6 (*Cercopithecus aethiops* derived epithelial kidney) cells were grown in Dulbecco’s modified Eagle’s medium (DMEM) supplemented with 2.5% heat-inactivated fetal calf serum (FCS), 2 mM l-glutamine, 100 units/mL penicillin, 100 μg/mL streptomycin, 1 mM sodium pyruvate, and non-essential amino acids (Sigma #M7145, St. Louis, MI, USA). For experiments in the presence of breast milk, the medium was supplemented with 100 µg/mL gentamicin. Cells were grown at 37 °C in a 5% CO_2_ humidified incubator.

### 2.2. Virus Strains and Virus Propagation

The African ZIKV strain MR766 was isolated in 1947 from a sentinel rhesus macaque [1]. Asian and pathogenic strains PRVABC59 or FB-GWUH-2016 were isolated in 2015 from a human serum specimen [36] or from a fetal brain with severe abnormalities [19], respectively. For virus propagation see [37]. In brief, 70% confluent Vero E6 cells in 175 cm²-cell culture flasks were inoculated with ZIKV in 5 ml medium for 2 h, before 40 mL medium was added. Cells were monitored for 3 to 5 days and supernatant was harvested when 70% of the cells detached due to cytopathic effects. Supernatants were centrifuged for 3 min at 325× *g* to remove cellular debris, and then aliquoted and stored at −80 °C as virus stocks. The infectious virus titer was determined by the tissue culture infectious dose 50 (TCID_50_) titration and the genome copy number of the stocks was assessed by RT-qPCR.

### 2.3. Breast Milk

Human breast milk was donated by healthy volunteering nursing mothers after informed consent had been signed. None of the mothers reported stays in ZIKV endemic regions or previous infections with flaviviruses. Samples were collected into sterile 50 mL Falcon tubes immediately before breastfeeding and either used for analysis within 30–60 min after donation or after long-term (>6 months) storage at −20 °C. All procedures were approved by the local ethics committee of Ulm University (337/18, 19 September 2018). 

### 2.4. Cell Viability Assay

The effect of human breast milk on the metabolic activity of the cells was analyzed using the CellTiter-Glo^®^ Luminescent Cell Viability Assay (Promega #G7571, Madison, WI, USA). It was performed according to the manufacturer’s instructions under conditions corresponding to the respective infection assays. Briefly, the medium was removed from the culture after 2 days of incubation and 50 µL phosphate-buffered saline (PBS) and 50 µL substrate reagent were added. After 10 min, the luminescence of the samples was measured in an Orion II Microplate Luminometer (Titertek Berthold, Pforzheim, Germany). Untreated controls were set to 100% viability.

### 2.5. TCID_50_ Endpoint Titration

To determine the tissue culture infectious dose 50 (TCID_50_), ZIKV stocks were serially diluted 10-fold and used to inoculate Vero E6 cells. To this end, 6000 Vero E6 cells were seeded per well in 96 flat-bottom well plates in 100 µL medium and incubated over night before 80 µL fresh medium was added. Next, 20 µL titrated ZIKV of each dilution was used for inoculation, resulting in final ZIKV dilutions of 1:10^1^ to 1:10^9^ on the cells in triplicates. Cells were then incubated for at least 6 days and monitored for cytopathic effect. TCID_50_/mL was calculated according to Reed and Muench.

### 2.6. Cell-Based ZIKV Immunodetection Assay

For infection, 6000 Vero E6 cells were seeded in flat-bottom 96-well plates in 100 µL medium, incubated overnight, and then topped to 180 µL medium. ZIKV stocks were diluted to the desired multiplicity of infection (MOI). In the “virion treatment” setting, virus was then mixed (1:1) with serially titrated breast milk samples and incubated for 10 min at room temperature. Then, cells were inoculated with 20 µL of the mixture in triplicates (resulting in a 10-fold lower concentration of the milk samples) and incubated for 2 h before the medium was replaced by fresh medium. Alternatively, cells were first incubated with the breast milk samples for 10 min, followed by ZIKV inoculation (“cell treatment”). Two days post infection, the flavivirus E protein content in the cells was quantified by cell-based immunodetection assay as described [38]. To this end, the medium was removed, and the plates were washed with PBS once before fixing the cells with 4% paraformaldehyde (PFA) for 20 min at room temperature. After aspirating PFA, cells were permeabilized by incubation with 100% ice cold methanol for 5 min, and again washed with PBS. Then, cells were incubated with the mouse anti-flavivirus protein E antibody 4G2 (Absolute Antibody #Ab00230-2.0, Redcar, Cleveland, UK) in antibody buffer (PBS containing 10% (*v*/*v*) FCS and 0.3% (*v*/*v*) Tween 20) at 37 °C. After 1 h, the cells were washed three times with washing buffer (0.3% (*v*/*v*) Tween 20 in PBS) before a secondary anti-mouse antibody conjugated with horseradish peroxidase (HRP) (Thermo Fisher #A16066, Waltham, MA, USA) was added and incubated for 1 h at 37 °C. Following four times of washing, the TMB peroxidase substrate (Medac #52-00-04, Wedel, Germany) was added and the reaction was stopped with 0.5 M H_2_SO_4_ after 5 min of light-protected room temperature incubation. The optical density (OD) was recorded at 450 nm and baseline corrected for 650 nm using the VMax Kinetic ELISA microplate reader (Molecular Devices, San José, CA, USA). 

### 2.7. Flow Cytometry

For infection, 100,000 Vero E6 cells were seeded in 24 well plates in 500 µL medium, incubated overnight, and then replaced by 180 µL fresh medium. Cells were then inoculated with 20 µL ZIKV that was treated for 10 min with thawed human milk. After 2 h, the inoculum was removed, cells were washed with PBS and fresh medium was added. Three days post infection, cells were detached (0.05% trypsin in PBS) and flow cytometric analysis was performed to determine infection rates. Specifically, cells were washed with PBS and stained with fixable viability stain 450 (BD Biosciences, San Jose, CA, USA) for 15 min at room temperature, followed by a washing step. Next, cells were fixed with 4% PFA for 10 min. Cells were then washed with fluorescence-activated cell sorting (FACS) buffer (1% (*v*/*v*) FCS in PBS) and permeabilized for 5 min with permeabilization solution (300 mM sucrose, 3 mM MgCl_2_, 50 mM NaCl, 20 mM Tris, 0.5% Triton X-100 in H_2_O). Cells were washed with 0.1% (*v*/*v*) Tween 20 in PBS and incubated with mouse anti-flavivirus protein E antibody 4G2 (in PBS containing 0.1% (*v*/*v*) Tween 20) for 30 min at room temperature. After a next washing step, cells were incubated with a secondary anti-mouse antibody conjugated with Alexa Fluor 488 (Thermo Fisher #A11001, Waltham, MA, USA) for 30 min at room temperature protected from light. After further washing steps, cells were resuspended in FACS buffer and analyzed in a BD FACS Canto™ II Cell Analyzer and evaluated with BD FACS Diva™ and FlowJo 7 software (BD Biosciences, San Jose, CA, USA).

### 2.8. Breast Milk Fractionation

The fractionation of human breast milk was performed as previously described [35]. Briefly, 10 mL pooled thawed breast milk was centrifuged for 20 min at 10,000–20,000× *g* at 4 °C to skim the fat (cream) from milk. Next, cream was resuspended in 10 mL PBS and the skim milk was ultracentrifuged at 100,000× *g* for 90 min at 4 °C to separate caseins from serum proteins. The casein-containing pellet was resuspended in 10 mL PBS and the supernatant (whey) was collected.

### 2.9. RT-qPCR Detection of Viral RNA of Intact ZIKV Particles

To determine the amount of intact ZIKV virions and their destruction by breast milk or detergents, the total and virion-associated viral RNA was determined by qPCR, as previously described for HCV [35]. To this end, a ZIKV MR766 stock containing ~10^10^ genomic RNA copies/mL was incubated with 90% PBS, 0.5% SDS, 1% Triton X-100, or 90% thawed breast milk for 1 h at 37 °C. Next, the free (and not virion protected) viral RNA of one aliquot of each sample was degraded by incubation with 10 U RNaseA (Qiagen, Hilden, Germany) for 1 h at 37 °C. As control (determining the total RNA), one aliquot was in parallel incubated with PBS. After treatment, the samples were analyzed for remaining RNA copy numbers. Quantitative real-time PCR was performed using qScript™ XLT One-Step-RT-qPCR ToughMix^®^ (Quanta Biosciences, Beverly, MA, USA). The final 20 µL RT-qPCR reaction mixture contained 5 µL RNA template, 10 pmol forward primer ZIKV-RKI-F (5’-ACGGCYCTYGCTGGAGC-3’, biomers.net, Ulm, Germany), 10 pmol reverse primer ZIKV-RKI-R (5’-GGAATATGACACRCCCTTCAAYCTAAG-3’; biomers.net, Ulm, Germany) and 3 pmol probe ZIKV-RKI-P (Fam-5’-AGGCTGAGATGGATGGTGCAAAGGG-3’-BMN-Q535; biomers.net, Ulm, Germany). The amplification was done under the following conditions, 10 min at 50 °C for RT reaction, 10 min at 95 °C for polymerase activation, followed by 40 cycles for 15 s at 95 °C, 60 s at 55 °C and 10 s at 72 °C.

### 2.10. Statistical Analysis

The determination of the inhibitory concentration 50 (IC_50_) by four-parametric nonlinear regression was performed using GraphPad Prism version 8.1.0 for Windows, GraphPad Software, San Diego, California USA, www.graphpad.com.

## 3. Results

### 3.1. Stored Human Breast Milk Is a Potent Inhibitor of ZIKV Infection

As ZIKV is shed into human breast milk [15,20,21,22,23,24,25,28,29], we analyzed the effect of this body fluid on ZIKV infection. For this, three virus strains were used: the prototype African MR766 strain isolated from a rhesus macaque [1], the more recent and Asian lineage derived strains FB-GWUH-2016 (GWUH) which was isolated from a fetal brain with severe abnormalities [19] and PRVABC59 (PRV), which was obtained from a human serum specimen [36]. TCID_50_-adjusted amounts of virus were mixed with equal volumes of serially diluted pooled breast milk. After incubation for 10 min, the mixtures were used to inoculate Vero E6 cells, resulting in a 10-fold dilution of the virus/milk mixture. After 2 h, the inoculum was replaced with fresh medium, and ZIKV infection rates were quantified 2 days later by a ZIKV E antigen specific cell-based immunodetection assay. To detect possible confounding cytotoxic effects of the breast milk, the same experiments were performed in the absence of virus and, after 2 days, intracellular ATP levels were determined by CellTiter-Glo Assay.

Initially, we used breast milk that was pooled from long-term frozen samples of four individual donors. Strikingly, thawed milk efficiently inhibited infection by the three ZIKV isolates with average IC_50_s of only 0.46 ± 0.03% milk during virion treatment (Figure 1a, Appendix Aa). The complete inhibition of ZIKV infectivity was achieved by only 1–2% of milk during virion exposure (Figure 1a, Appendix Aa), corresponding to final cell culture concentrations of 0.1–0.2% milk, which is far below the cytotoxic concentration (Figure 1b). To exclude that milk proteins may interfere with viral quantification by the enzymatic readout of the cell-based immunodetection assay, we also applied flow cytometry of ZIKV-infected cells. As shown (Appendix Ab,c), concentrations of ≥0.5% of milk reduced the percentage of infected cells from 82% to below 1%. Similar results were obtained when analyzing the mean fluorescence intensities of the cultures (Appendix Ad). The anti-ZIKV activity of breast milk was confirmed with freeze-stored samples derived from four independent nursing mothers (Figure 1c). With little variation in the antiviral activity, all four samples completely inhibited viral infection at concentrations exceeding 0.3% during virion treatment (Figure 1c).

### 3.2. The Anti-ZIKV Factor in Milk Destroys the Structural Integrity of the Virion

To test whether milk acts on the virus or the cell, ZIKV was incubated with up to 20% of pooled breast milk for 10 min (“virion treatment”) and then added to cells, resulting in a 10-fold dilution of the inoculum, and final cell culture concentrations of up to 2% of milk. Simultaneously, Vero E6 cells were first incubated with breast milk at concentrations of up to 2% for 10 min, before ZIKV was added (“cell treatment”). Thus, in both treatments the final milk concentrations in cell culture are the same, however, due to virion pre-exposure with 10-fold higher concentrations different outcomes could arise if the antiviral factors target the virion. After 2 h, the supernatants were exchanged with fresh medium, and viral infection was quantified 2 days later. Under both conditions, milk inhibited ZIKV infection (Figure 2a). However, milk concentrations during virion treatment determined the magnitude of virus inhibition (IC_50_ virion: 0.27 ± 0.04%; IC_50_ cell: 1.02 ± 0.09%), showing that the antiviral activity in milk is directed against the virus. We next analyzed whether milk may directly disrupt virion integrity, applying a protocol established for HCV [35]. In brief, virions were exposed to PBS, detergents SDS or Triton X-100, or pooled (stored) milk for one hour at 37 °C. Thereafter, a ribonuclease was added to degrade free genomic viral RNA released from destroyed virions. Next, ZIKV RNA (total and remaining) was quantified by RT-qPCR. As shown in Figure 2b, comparable numbers of viral genome copies per mL (10^8^ to 10^9^) were detected in all control samples, demonstrating that detergents or milk components do not interfere with RT-qPCR-based detection of viral genomes. RNase treatment, however, resulted in an almost entire loss of viral RNA when virions were exposed to breast milk (Figure 2b). Similarly, a more than 99% reduction in viral genome copies was determined upon detergent treatment. Next, we analyzed how fast ZIKV is inactivated by milk. We found that incubating ZIKV with 0.5% milk for only one minute did not reduce viral infectivity (Figure 2c). However, infectivity decreased in a time-dependent manner and was lost entirely after 30 min of incubation (Figure 2c). These data show that the antiviral factor in breast milk interacts with Zika virions in a way that results in the rapid release of genomic RNA, which can only be achieved by physical destruction of the viral particles, explaining loss of infectivity.

### 3.3. The ZIKV Inhibitory Factor is Present in the Cream Fraction of Milk

To clarify which component in milk is responsible for the observed anti-ZIKV activity, pooled long-term stored milk was separated by low speed centrifugation into cream and skim milk, as described [35]. Skim milk was further fractionated by high-speed centrifugation into the pellet containing caseins and into the supernatant called whey [39]. None of the milk fractions reduced metabolic activity of Vero E6 cells at final cell culture concentrations of up to 2% (*v*/*v*) (Figure 3a). We tested the effect of the milk fractions on ZIKV infection and found that only the cream fraction led to a reduced ZIKV infectivity (IC_50_ of 0.85 ± 0.15% during virion treatment) that was comparable to whole milk (IC_50_ of 0.41 ± 0.03%) (Figure 3b). Skim milk and the whey fraction displayed some antiviral activity (IC_50_ of 13.12 ± 1.31 and 15.23 ± 1.19%, respectively) whereas the casein fraction had no effect (Figure 3b).

### 3.4. Fresh Milk Does Not Inhibit ZIKV Infection but Becomes Antivirally Active in a Time-Dependent Manner

Our finding that milk has potent anti-ZIKV activity (Figure 1, Figure 2 and Figure 3) was obtained with human breast milk that was stored and frozen at −20 °C for prolonged periods of time. To corroborate our results with fresh samples, milk was obtained from three nursing mothers and analyzed within 30 min after donation. Conversely, none of the fresh milk samples inhibited ZIKV, even at concentrations of up 50% during virion treatment (Figure 4). We next studied whether storage of these fresh milk samples may result in a gain of anti-ZIKV activity. In fact, upon incubation at 37 °C for several hours, all milk samples reduced ZIKV infection in a dose-dependent manner (Figure 5a). We observed some donor variation, e.g., the milk of donor 1 and 2 became antivirally active already after 1–2 h whereas the milk of donor 3 inhibited ZIKV only when incubated for 8 hours or longer (Figure 5a). Similar results were obtained if milk was incubated at 22 °C (Figure 5b). We were then wondering how storage of milk at 4 °C would impact antiviral activity. Milk derived from donor 1 and 2 became slightly more antivirally active as compared to the 37 °C condition, however for donor 3 we observed an opposite effect (Figure 5c). Finally, freezing milk at −20 °C resulted in slightly increased antiviral activities of donor 1, but not donor 2 and 3 (Figure 5d). Of note, none of the milk samples displayed cytotoxic effects at the concentrations and conditions tested, except for donor 2 at the highest dose upon 37 °C incubation for 8 h (Appendix A). Thus, fresh human milk does not affect ZIKV infection but its storage results in a donor- and time-dependent generation of factor(s) that inactivate ZIKV.

### 3.5. Loss of ZIKV Infectivity in Milk Stored at 4 °C

We next analyzed the stability of the virus in milk during storage. Pfaender et al., analyzed storage of ZIKV in milk at 4 °C and pasteurization at 63 °C [34]. Here, we explored storage of milk at physiological and common temperatures of 37 °C and 22 °C. For this, freshly donated milk (or PBS as control) was spiked with ZIKV MR766, GWUH or PRV, and incubated for up to 8 h at 37 °C (Figure 6a) or 22 °C (Figure 6b). Viral infectivity was determined at indicated time points by TCID_50_ analysis showing that ZIKV that was incubated in PBS remained fully infectious, even after incubation for 8 days (Figure 6). Similarly, the incubation of ZIKV in milk at 22 °C did not affect ZIKV infectivity (Figure 6b). In contrast, the incubation of ZIKV in milk for 8 h at 37 °C resulted in a 97% or 99% reduction in viral titers for donor 4 or 5, respectively (Figure 6a, MR766). Similarly, titers of GWUH or PRV were reduced by 99% or 44% (Figure 6a, GWUH) and 99% or 90% (Figure 6a, PRV) for the two donors. These data show that under physiological conditions of 37 °C, ZIKV may be inactivated in breast milk. 

Next, we assessed whether storage of ZIKV contaminated milk in the fridge or the freezer may allow virus inactivation. Thus, the three ZIKV strains were spiked into fresh milk or buffer and incubated for up to 10 days at 4 °C (Figure 6c) or −20 °C (Figure 6d). TCID_50_ analysis revealed that incubation at 4 °C for 2 days led to an entire loss of ZIKV infectivity in both milk samples (Figure 6c), confirming previous data [34]. Incubation at −20 °C, however, had not significant effects on viral titers (Figure 6d). Thus, storage of milk from a ZIKV infected mother at 4 °C but not at −20 °C reduces viral titers over time and should thus limit ZIKV transmission by breast milk feeding.

## 4. Discussion

Besides ZIKV transmission through bites of infected mosquitoes, the virus may also be transmitted directly between individuals by exchange of contaminated body fluids, e.g., during sexual intercourse [14,15,16,17], blood transfusions [15,18], giving birth [15,20], or possibly breastfeeding [15,20,21,22,23,24,25]. Viral RNA is detectable in the milk of ZIKV-infected mothers [15,20,21,22,23,24,25,28,29] and the infectious virus could be isolated from breast milk [15,21,22,23,24,25,28,29], questioning the safety of breastfeeding the infant. Thus, here, we analyzed the effect of milk on viral infection. 

We found that fresh human breast milk obtained from individual donors does not affect ZIKV infection (Figure 4), which was somewhat surprising given the numerous reports on antimicrobial activity of milk published so far [30,31,33,35,40,41,42,43,44]. However, the same milk samples gained potent anti-ZIKV activity upon storage for 0.5–10 h (Figure 5). Anti-ZIKV activity varied greatly between donors and was affected by temperature and incubation times (Figure 5). Some samples became antivirally active as quickly as after 30 min (Figure 5c+d, donor 1+2), whereas others hardly inhibited infection after 10 h of incubation (Figure 5d, donor 3). Overall, the antiviral activity increased in a time-dependent manner, suggesting that the antiviral factor is generated over time. Surprisingly, however, after 10 h of incubation, IC_50_ values were not as low as for milk that was stored for more than 6 months at −20 °C (Figure 1a). This suggests that a continuous incubation might further increase milk’s antiviral activity, probably until a maximum activity is reached.

What is the nature of the anti-ZIKV factor in milk and how is it generated? We can exclude that ZIKV neutralizing antibodies in milk are responsible as the fresh milk samples did not show anti-ZIKV activity (Figure 4). Additionally, the fact that the antiviral factor is generated during incubation at all tested temperatures suggests that it is also not of bacterial origin as bacteria do not grow and produce toxins at −20 °C. We found that the antiviral activity in milk is directed against the ZIKV particle (Figure 2a) and results in a rapid destruction of viral particle integrity (Figure 2b,c). Furthermore, the fat containing cream fraction of milk contains the antiviral factor (Figure 3b). These findings are very similar to those obtained by Pfaender et al. on HCV, another member of the *Flaviviridae* family. In this study, the authors show that lipases in milk generate free fatty acids that destroy the membrane of HCV particles [35]. This is a common antiviral property of fatty acids [45] and their release and antiviral activity in milk has been demonstrated for enveloped viruses of different families [46,47]. Mechanistically, it has been discussed that the free fatty acids likely form micellar structures and interact with or are even incorporated into the viral membrane, thus destabilizing it [35,46]. The two lipases in human breast milk are the lipoprotein lipase (LPL) and the bile salt-stimulate lipase (BSSL), which are associated with the casein fraction [48,49,50]. It is well-described for bovine and has also been observed for human milk that upon cooling at 4 °C, LPL migrates from the casein to the cream fraction, resulting in increased free fatty acid release (“cooling activation”) [50,51,52,53]. Furthermore, the milk-fat-globule membrane that protects lipids from direct lipolysis by LPL can be ruptured by agitation, foaming (and likely also vortexing), or freezing [54,55,56] allowing access for LPL to the core lipids, subsequent lipolysis and generation of antivirally active fatty acids. Similarly, it has been reported that BSSL, typically dependent on bile salts in the infant’s stomach, is also activated after freezing [50,57,58]. Thus, collectively these data suggest that lipases released or activated during processing of milk samples are responsible for the generation of free fatty acids that destroy the ZIKV membrane. Moreover, the fat composition of milk also differs between and within individuals, which may explain the observed donor variations [59,60]. Pfaender et al. proposed that in addition to bile salt BSSL activation in the infant’s stomach [57,58], gastric lipases may also release antivirally active fatty acids upon breast feeding [35]. Especially in combination, this may result in high levels of antiviral fatty acids and thus inactivation of ZIKV in the infant’s stomach and subsequently reduced rates of transmission. This is in line with observations that antiviral fatty acids are found in infants’ stomachs one hour after breast feeding [47]. However, further studies are required to clarify the exact mechanism underlying the anti-ZIKV activity of human breast milk and to examine its antiviral potency against other enveloped viruses. 

Besides determining the effect of milk on ZIKV infection, we also analyzed how milk affects ZIKV infectivity that was spiked into the body fluid, a setting which more closely reflects the in vivo situation. We observed a time- and donor-dependent reduction of viral titers by 1–2 orders of magnitude if milk/virus was incubated for more than 2 h at 37 °C, suggesting that similar rates of ZIKV inactivation may also occur in the mother. Incubation at 22 °C for the same period of time had no effect on viral titers (Figure 6b), perhaps because lipases or other enzymes involved in the generation of the antiviral factor are inactive at this temperature. We also analyzed how storage of ZIKV contaminated milk in the fridge (4 °C) or freezer (−20 °C) affects viral titers. Interestingly, incubation at 4 °C for 2 days resulted in an entire loss of infectivity of the three ZIKV strains tested (Figure 6c), with the exception of a small “blip” just above the detection limit observed for donor 5 at day 8 which disappeared again at day 10. These data confirm recent findings by Pfaender et al. showing effective inactivation of the African ZIKV strain MP-7051 and the Puerto Rican strain PRVABC59 in human breast milk under the same conditions [34] and might be explained by the “cooling activation” of the lipases as already discussed [51,52,53]. Surprisingly, the incubation of the samples at −20 °C did not result in any loss of infectivity over time (Figure 6d). This seems to be in contrast to the observations made above by mixing virus with milk that had been incubated before (Figure 1, Figure 2, Figure 3, Figure 4 and Figure 5). At −20 °C the integrity of micelles is damaged and the BSSL activated, resulting in lipases getting in contact to a large amount of lipids upon thawing. As observed in Figure 2c, long-term frozen milk inactivates ZIKV within 10–30 min. When performing experiments with milk that had been frozen up to 10 h (Figure 5d) or long-term (Figure 1a), this was the time frame of incubation with ZIKV before inoculation of cells. In the spiking setup (Figure 6), ~2–3 log higher viral titers were mixed with milk, frozen for indicated time points, then thawed and immediately titrated and added to the cells, resulting in maximum concentrations of only 0.09% milk in cell culture (Figure 6d). Thus, the low milk concentration and short incubation time at temperatures higher than −20 °C might not have been sufficient to inactivate high titers of ZIKV. In future studies it would be interesting to clarify whether the antiviral factor is generated at −20 °C but exerts its antiviral activity at warmer temperatures. Such experiments might also help to confirm that the antiviral factor can be generated and be active at 4 °C (Figure 6c), and why this effect is not seen at 22 °C and only marginally at 37 °C. 

Pfaender et al. have demonstrated that Holder pasteurization (62.5 °C for 30 min) of ZIKV-contaminated milk effectively destroys viral infectivity [34]. Although pasteurization is effective, it may not be applicable under resource-poor settings. Our results and those by Pfaender et al. show that storage of milk from ZIKV-infected mothers in the fridge for at least 2 days may in the majority of cases reduce or even eliminate risk of ZIKV transmission to the infant. Thus, storage of milk in the fridge might provide an interesting alternative to pasteurization. However, possible donor variations and the limited sample size tested preclude making generalized recommendations. 

Recent reviews investigated transmission from ZIKV-infected mothers to their children and did not find evidence for efficient ZIKV transmission via breastfeeding, despite the presence of virus in breast milk [15,24,25] and three reports of transmission [20,22,23,25]. This is in line with the findings in a mouse model, where infectious ZIKV was detected in breast milk but did not result in transmission to pups [61]. The finding by others and us that ZIKV is inactivated in milk before and after feeding provides a plausible explanation for the rare event of transmission by breastfeeding.

## Figures and Tables

**Figure 1 viruses-11-00591-f001:**
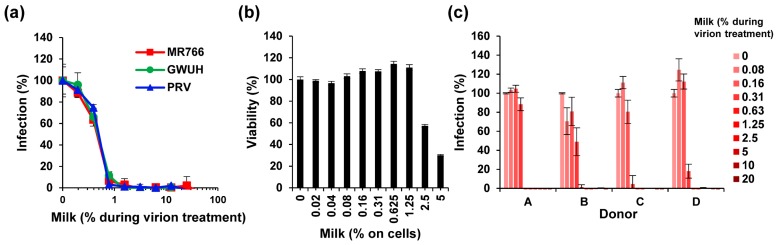
Effect of long-term frozen breast milk on ZIKV infection. (**a**) ZIKV MR766, GWUH or PRV (MR766: 1.58, GWUH: 2.51, PRV: 1.58 × 10^7^ TCID_50_/mL) were mixed 1:1 with thawed breast milk at indicated concentrations and incubated for 10 min at room temperature. Milk was stored for more than 6 months at −20 °C. ZIKV/milk mixtures were then used to inoculate Vero E6 cells resulting in a 10-fold dilution of the samples. Then, 2 h later, the medium was changed and, 2 days later, the infection rates were determined by a cell-based immunodetection assay that enzymatically quantifies the flavivirus protein E. (**b**) Vero E6 cells were incubated with thawed pooled breast milk at indicated concentrations for 2 h. The medium was then replaced, and the cellular viability was determined 2 days later by CellTiter-Glo Assay. Data are normalized to viability in the absence of milk. (**c**) Thawed breast milk from four individual donors was incubated with ZIKV MR766 (1.58 × 10^7^ TCID_50_/mL) at indicated concentrations before inoculation of Vero E6 cells as described in (**a**). Infection data are normalized to infection rates in the absence of the respective sample. Data are represented as average values obtained from triplicate infections ± standard deviations.

**Figure 2 viruses-11-00591-f002:**
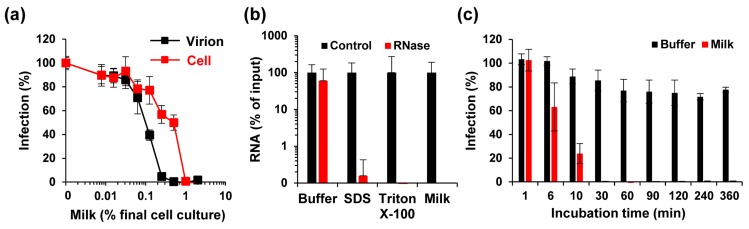
Mechanism of ZIKV inhibition by long-term freeze-stored breast milk. (**a**) For cell treatment (cell), Vero E6 cells were incubated with indicated concentrations of breast milk, and thereafter inoculated with ZIKV MR766. For virion treatment (virion), ZIKV MR766 (1.58 × 10^7^ TCID_50_/mL) was incubated 1:1 for 10 min with 10-fold higher breast milk concentrations before the mix was diluted onto Vero E6 cells. Concentrations shown correspond to the final concentrations of milk (*v*/*v*) in cell culture. Then, 2 h after inoculation, the medium was changed and, 2 days later, the infection rates were determined by a cell-based immunodetection assay that enzymatically quantifies the flavivirus protein E. (**b**) ZIKV MR766 was incubated with PBS (90%), SDS (0.5%), Triton X-100 (1%) or milk (90%) for 1 h at 37 °C. Free (and released) RNA was incubated in buffer or degraded by RNase A (10 U) for 1 h at 37 °C. Remaining ZIKV RNA copy numbers were determined by RT-qPCR. (**c**) PBS or breast milk (5%) were incubated with ZIKV MR766 for the indicated time at room temperature before the mixture was inoculated onto Vero E6 cells. After 2 h, the medium was changed and, 2 days later, the infection rates were determined as described in (**a**). Infection data are normalized to infection rates in the absence of the respective sample. All data are represented as average values obtained from triplicates ± standard deviations.

**Figure 3 viruses-11-00591-f003:**
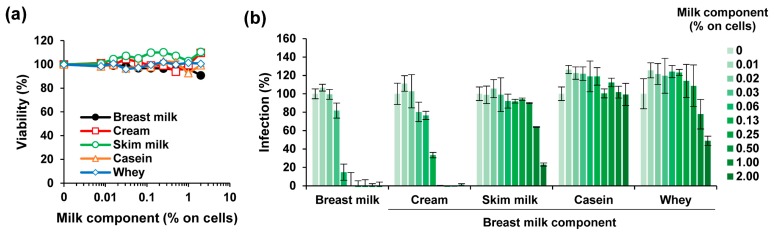
Effect of fractions derived from long-term freeze-stored breast milk on ZIKV infection. (**a**) Pooled long-term stored breast milk was defatted by centrifugation, the cream collected and resuspended in PBS at the original volume. The remaining skim milk was further ultracentrifuged to separate caseins from serum proteins. The casein-containing pellet was resuspended in PBS to the original volume and the supernatant (whey) collected. The breast milk and its components were then incubated on Vero E6 cells at indicated concentrations for 2 h. The medium was then replaced, and the cellular viability was determined 2 days later by CellTiter-Glo Assay. (**b**) Breast milk and its components were mixed 1:1 with ZIKV MR766 (1.58 × 10^7^ TCID_50_/mL) for 10 min before 10-fold dilution onto Vero E6 cells resulting in indicated concentrations. Then, 2 h after inoculation, the medium was changed and, 2 days later, the infection rates were determined by a cell-based immunodetection assay that enzymatically quantifies the flavivirus protein E. All data are normalized to values in the absence of the respective sample. Data are represented as average values obtained from triplicate infections ± standard deviations.

**Figure 4 viruses-11-00591-f004:**
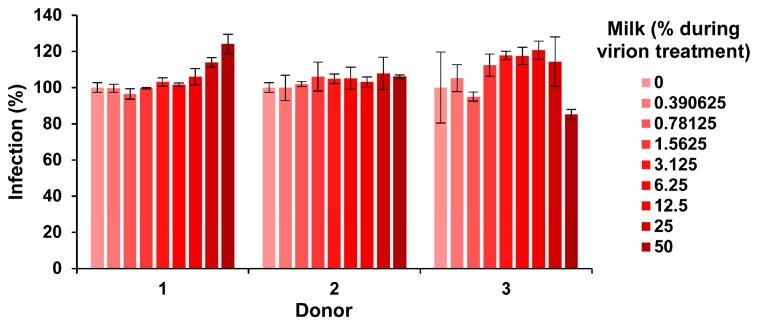
Effect of fresh milk on ZIKV infection. Fresh breast milk was received from three donors and, after 30 min, was mixed 1:1 with ZIKV MR766 (1.58 × 10^7^ TCID_50_/mL) at indicated concentrations and incubated for 10 min at room temperature. Mixtures were then diluted 10-fold onto Vero E6 cells and incubated for 2 h before the medium was changed. Infection rates were determined 2 days later by an E protein immunodetection assay. Infection data are normalized to infection rates in the absence of the respective breast milk sample and represent average values obtained from triplicate infections ± standard deviations.

**Figure 5 viruses-11-00591-f005:**
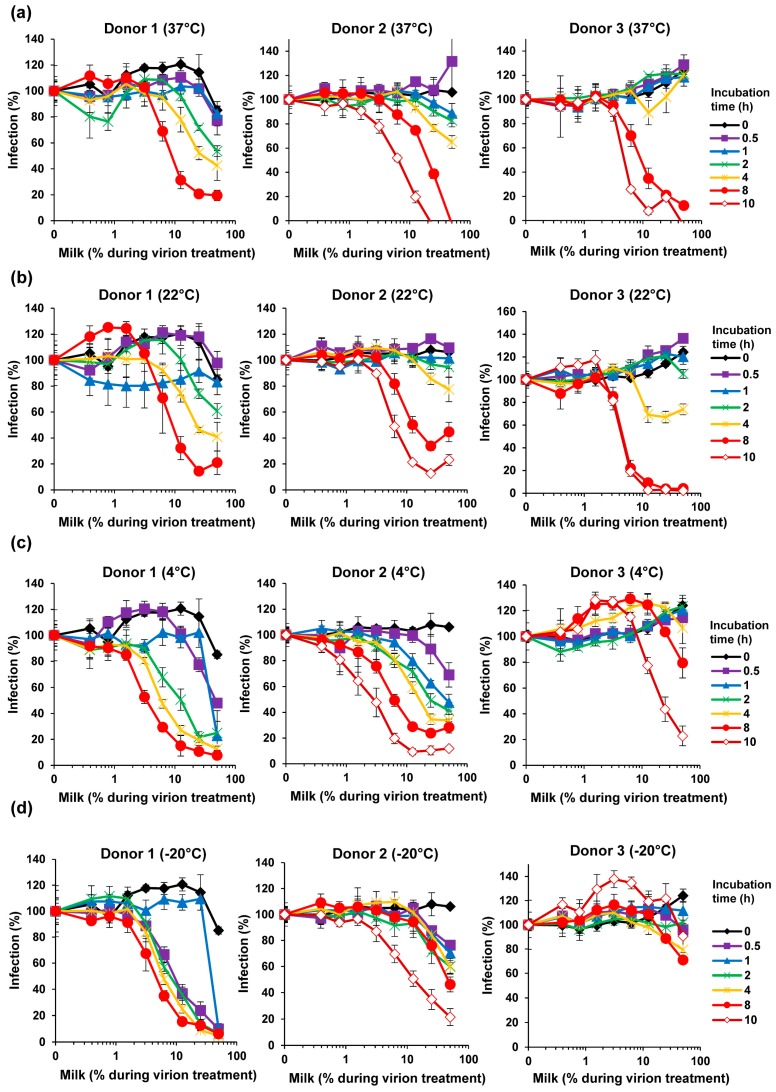
Effect of storage and temperature on the anti-ZIKV activity of milk. Fresh breast milk was received from three donors and, after 30 min, was mixed 1:1 with ZIKV MR766 (1.58 × 10^7^ TCID_50_/mL) at indicated concentrations and incubated for 10 min at room temperature. Mixtures were then diluted 10-fold onto Vero E6 cells and incubated for 2 h before the medium was changed. Additionally, the breast milk was incubated at (**a**) 37 °C, (**b**) 22 °C, (**c**) 4 °C, or (**d**) −20 °C for indicated time points before mixing with ZIKV and inoculation of Vero E6 cells. Infection rates were determined 2 days later by a cell-based immunodetection assay that enzymatically quantifies the flavivirus protein E. Infection data are normalized to infection rates in the absence of the respective breast milk sample and represent average values obtained from triplicate infections ± standard deviations. See also Appendix A.

**Figure 6 viruses-11-00591-f006:**
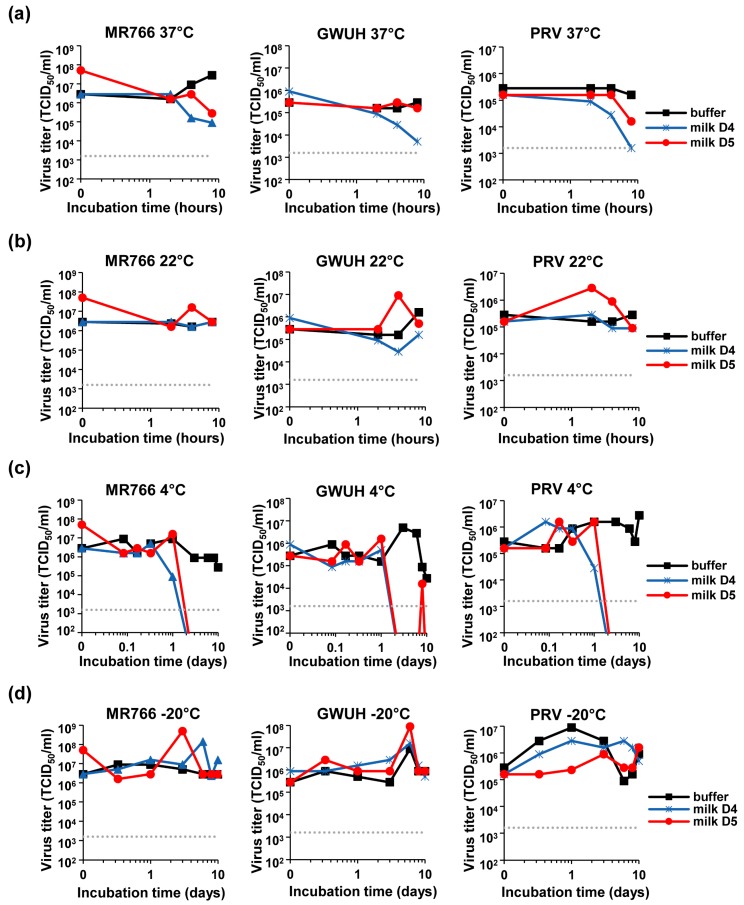
Determination of the ZIKV titer after the incubation of the virions in breast milk at different temperatures. ZIKV MR766, GWUH, or PRV were spiked into buffer or fresh breast milk that was received from two donors resulting in 90% buffer or milk, respectively. Samples were then incubated at (**a**) 22 °C, (**b**) 37 °C, (**c**) 4 °C or (**d**) −20 °C for indicated time points up to 8 h (**a**+**b**) or 10 days (**c**+**d**), before the virus titer was determined by TCID_50_ titration onto Vero E6 cells.

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
