# Peer review of "Storage-Dependent Generation of Potent Anti-ZIKV Activity in Human Breast Milk"

_viruses, 2019, doi:10.3390/v11070591_

Round 1
Reviewer 1 Report
only one suggestion to Introduction:
1. "This factor rapidly abrogates infectivity by physical destruction of the viral particle and may play a role in virus inactivation upon breast feeding in the gastrointestinal tract of babies or upon storage of milk for later use. This might explain why ZIKV transmission via breast feeding is hardly observed."
the authors stated in prior sentences that ZKIV fresh milk has no effects on ZIKV, infants on breast feeding from mother is having the fresh milk, so how it explains why the ZIKV transmission via breast feeding is hardly observed?
Author Response
1. "This factor rapidly abrogates infectivity by physical destruction of the viral particle and may play a role in virus inactivation upon breast feeding in the gastrointestinal tract of babies or upon storage of milk for later use. This might explain why ZIKV transmission via breast feeding is hardly observed."
the authors stated in prior sentences that ZKIV fresh milk has no effects on ZIKV, infants on breast feeding from mother is having the fresh milk, so how it explains why the ZIKV transmission via breast feeding is hardly observed?
Response 1: Thank you for pointing out the lack of coherence. We now clarified that our findings match those of Pfaender et al., that antivirally active fatty acids might be released by milk and gastric lipases in the stomach of babies. See lines 63-67.

Reviewer 2 Report
In this study, Conzelmann and colleagues demonstrate that human breast milk that has been stored possesses an anti-ZIKV activity. This is mediated by a disruption of the integrity of the viral particle. The study is generally well written and conducted. Results are clear and convincing and the conclusions fit with the data. This raises very interesting questions about the antiviral potential of breast milk. However, I think that the discussion would gain to be more elaborated.
First, considering that another similar study (despite much shorter and less elaborated than this one) was previously published by the group of Volker Thiel, I think that the authors should better highlight the novelty of their results in the discussion.
Second, it would be relevant to discuss a bit more how the milk impact on the integrity of the virion. Is it an effect on envelope protein conformation/stability, or membrane solubilization? The data from the RNA protection assays suggest an effect on the lipid membrane of the virus. The authors propose that this could be due to free fatty acids that are contained in the milk. Are they generally contained within the active “cream” fraction? Could the authors discuss more precisely this possible antiviral mode-of-action? Are there evidence in the literature that this would also work against other flaviviruses or RNA viruses?
Finally, although it is clear that breast milk alters the integrity of the virion, one cannot exclude that this body fluid also impacts on intracellular viral replication. Indeed, according to the data shown in Fig 2A, pretreatment of cells with milk decreases infection with an approximate EC50 of 0.3uM. I think that this should be discussed. Were the cells washed with PBS after the treatment to remove the milk before the infection? Indeed, residual milk could impact on the stability of the virus inoculum that is attached to the cell. In other words, do the authors believe that the “cell treatment” affect the stability of the virion interacting with the cell or rather intracellular steps of the life cycle? This should be at least discussed if not experimentally addressed.
Minor comments:
Line 52: Please change for “does not always correlate”
Line 363: HCV is not a flavivirus. It is a hepacivirus while both ZIKV and HCV both belong to the Flaviviridae family.
Author Response
Response to Reviewer 2 Comments
In this study, Conzelmann and colleagues demonstrate that human breast milk that has been stored possesses an anti-ZIKV activity. This is mediated by a disruption of the integrity of the viral particle. The study is generally well written and conducted. Results are clear and convincing and the conclusions fit with the data. This raises very interesting questions about the antiviral potential of breast milk. However, I think that the discussion would gain to be more elaborated.
First, considering that another similar study (despite much shorter and less elaborated than this one) was previously published by the group of Volker Thiel, I think that the authors should better highlight the novelty of their results in the discussion.
Response 1: We have rewritten the whole discussion and elaborately commented on how our observations can be explained by the generation and activity of the antiviral factor at different temperature to a degree that exceeds the earlier study.
Second, it would be relevant to discuss a bit more how the milk impact on the integrity of the virion. Is it an effect on envelope protein conformation/stability, or membrane solubilization? The data from the RNA protection assays suggest an effect on the lipid membrane of the virus. The authors propose that this could be due to free fatty acids that are contained in the milk. Are they generally contained within the active “cream” fraction? Could the authors discuss more precisely this possible antiviral mode-of-action? Are there evidence in the literature that this would also work against other flaviviruses or RNA viruses?
Response 2: We now discussed that our observations are in line with the findings that fatty acids released from the cream fraction of milk destroy viral membranes. We now cited the publications commenting on the mode-of-action and the potential general effect on enveloped viruses. See lines 369-372 + 389-391.
Finally, although it is clear that breast milk alters the integrity of the virion, one cannot exclude that this body fluid also impacts on intracellular viral replication. Indeed, according to the data shown in Fig 2A, pretreatment of cells with milk decreases infection with an approximate EC50 of 0.3uM. I think that this should be discussed. Were the cells washed with PBS after the treatment to remove the milk before the infection? Indeed, residual milk could impact on the stability of the virus inoculum that is attached to the cell. In other words, do the authors believe that the “cell treatment” affect the stability of the virion interacting with the cell or rather intracellular steps of the life cycle? This should be at least discussed if not experimentally addressed.
Response 3: We now explain the experimental setup and the rational of Fig 2A in more detail (lines 214-224). Milk was either 10-minute preincubated with virus, or on the cells. Without a washing step, the milk/virus mix was added to cells, or virus was added to the milk-exposed cells. The key difference in the experimental setting is that during the virion treatment the milk concentration is 10-fold higher than on the cells. If the milk acts on the virus particles it should therefore be more potent in the virion treatment setting - which was the case in our results. We do conclude that milk only affects the viral particles, and not any intracellular steps of viral replication.
Minor comments:
Line 52: Please change for “does not always correlate”
Response 4: Done.
Line 363: HCV is not a flavivirus. It is a hepacivirus while both ZIKV and HCV both belong to the Flaviviridae family.
Response 5: We corrected it to “member of the Flaviviridae family” (line 367)
Reviewer 3 Report
Viruses – Zika and breast milk
Conzelmann et al completed a study examining the effects of breast milk on Zika virus infectivity. The study is well written and describes interesting results demonstrating that frozen milk, particularly the fat portion, induces Zika virus particles to loose integrity and therefor the ability to infect. While interesting and well written, some data appears inconsistent and some of the conclusions are not well supported by the data.
While pooled frozen milk and frozen milk from 4 individual donors was very effective at killing ZIKV (Fig1), individual donor fresh milk had no activity (Fig 4). As with all data using diverse human materials, not all donor material behaved similarly. A second study taking the fresh donor milk and incubating it at different temperatures demonstrated that some time/temp combinations gained antiviral activity. While “aging” the milk increased antiviral activity in some scenarios (Fig 5), the individual data never reached the strong antiviral activity seen in the first figure. The discussion needs to highlight the variation found. For example in Fig 1, four different donor samples that were frozen eliminate viral infection at less than 2%. In Fig 4, three different donor samples are frozen and only 2 donors gain antiviral activity and that activity is not very potent (especially compared to Fig 1 results). How long was the milk frozen in Fig 1? Can you treat donor 1-3 milk in a similar manner and get similar results? Additional discussion summarizing the differences in the data would make the paper stronger.
Donor to donor variation is clearly present. I am concerned that some of the conclusions drawn from the study may be misinterpreted or understood by a lay audience. For example -Lines 353-354 – “storage of milk from ZIKV-infected mothers in the fridge for at least 2 days may reduce or even eliminate risk of ZIKV transmission to the infant.” This statement is based on data presented in Fig 6 which only looks at milk from two donors. While for two ZIKV strains, virus is undetectable after 2 days, one donor sample has virus present on day 10. There is no apparent time dependent pattern, for donor 5 there is no effect at day 1 and then all virus disappears at day 2. If there is some virucidal activity, what is happening between day 1 and 2 to make everything happen then?
Figure 2A – I am slightly confused about the design of this experiment. In figure 1 you demonstrate that Zika infection in the presence of 1% milk basically eliminates infection. In figure 2 you want to determine if the milk is interfering with the virus particles or altering the cells. However, the way it reads suggests that in the “cell” treatment, the milk was added to the cells and then the virus was added. In this scenario the virus and the milk still come in contact with one another. What if you treated the cells then removed the milk and infected? If the milk is causing viral particle integrity loss, I would predict simply treating cells with milk would have no antiviral activity.
Figure 2B – This is a nice experiment. I would suggest repeating with similar milk levels you use in part A. I am very surprised the milk was more effective than SDS and Triton. Showing dose dependence would greatly improve the conclusions one can draw from this experiment.
Figure 5 – Can you increase the size of the symbols in the legend. On my print out it was very difficult to determine which lines corresponded to which incubation time.
Expressed breast milk is not sterile and incubating at 37 degrees (and potentially 22 degrees) for 10 hours will cause bacterial growth. How can you conclude the antiviral activity is in the milk or a bi-product of bacterial growth/metabolites? Does the same thing happen when broad spectrum antibiotics are added to the milk to prevent bacterial growth?
How long was the milk frozen in Fig 1? Why does this milk not behave similarly?
In Fig 3 you show the cream portion contains most of the antiviral activity. Did the donor samples have similar fat content?
Fig 6 – Here you show virus is killed when kept in the milk at 4 degrees, but not in the freezer. Please discuss why the frozen milk does not do anything now?
Line 17 – not sure it is fair to claim Zika causes fatal complications in adults
Line 153 – change to Qiagen
Author Response
Response to Reviewer 3 Comments
Conzelmann et al completed a study examining the effects of breast milk on Zika virus infectivity. The study is well written and describes interesting results demonstrating that frozen milk, particularly the fat portion, induces Zika virus particles to loose integrity and therefor the ability to infect. While interesting and well written, some data appears inconsistent and some of the conclusions are not well supported by the data.
Point 1
While pooled frozen milk and frozen milk from 4 individual donors was very effective at killing ZIKV (Fig1), individual donor fresh milk had no activity (Fig 4). As with all data using diverse human materials, not all donor material behaved similarly. A second study taking the fresh donor milk and incubating it at different temperatures demonstrated that some time/temp combinations gained antiviral activity. While “aging” the milk increased antiviral activity in some scenarios (Fig 5), the individual data never reached the strong antiviral activity seen in the first figure. The discussion needs to highlight the variation found. For example in Fig 1, four different donor samples that were frozen eliminate viral infection at less than 2%. In Fig 4, three different donor samples are frozen and only 2 donors gain antiviral activity and that activity is not very potent (especially compared to Fig 1 results). How long was the milk frozen in Fig 1? Can you treat donor 1-3 milk in a similar manner and get similar results? Additional discussion summarizing the differences in the data would make the paper stronger.
Response 1: We now address this issue in our discussion and mention in the methods and results section how long the original frozen milk was stored (see lines 91 + 203 + 358). The milk used in Figs 1-3 was stored > 6 months at -20°C. As shown in Fig 5, at a maximum incubation time of 10 hours at -20°C, 50% milk completely abrogates viral infectivity. We therefore speculate that an incubation time over several months would increase the antiviral potency of milk as observed in Fig. 1 (see lines 355-359). We now discussed how the storage at -20°C affects the generation of the antiviral factor (lines 376-381).
Point 2
Donor to donor variation is clearly present. I am concerned that some of the conclusions drawn from the study may be misinterpreted or understood by a lay audience. For example -lines 353-354 – “storage of milk from ZIKV-infected mothers in the fridge for at least 2 days may reduce or even eliminate risk of ZIKV transmission to the infant.” This statement is based on data presented in Fig 6 which only looks at milk from two donors. While for two ZIKV strains, virus is undetectable after 2 days, one donor sample has virus present on day 10. There is no apparent time dependent pattern, for donor 5 there is no effect at day 1 and then all virus disappears at day 2. If there is some virucidal activity, what is happening between day 1 and 2 to make everything happen then?
Response 2: We agree with the reviewer and cautioned our statement to “that storage of milk from ZIKV-infected mothers in the fridge for at least 2 days may in the majority of cases reduce or even eliminate risk of ZIKV transmission to the infant. … However, possible donor variations and the limited sample size tested preclude to make generalized recommendations.” (lines 423-426). In the discussion we have now elaborated on how the antiviral factor is generated in milk (lines 366-383), and interpreted how this is in line with our observations (lines 394-419).
Point 3
Figure 2A – I am slightly confused about the design of this experiment. In figure 1 you demonstrate that Zika infection in the presence of 1% milk basically eliminates infection. In figure 2 you want to determine if the milk is interfering with the virus particles or altering the cells. However, the way it reads suggests that in the “cell” treatment, the milk was added to the cells and then the virus was added. In this scenario the virus and the milk still come in contact with one another. What if you treated the cells then removed the milk and infected? If the milk is causing viral particle integrity loss, I would predict simply treating cells with milk would have no antiviral activity.
Response 3: We now explain the experimental setup and the rational in more detail (lines 214-224). The key difference in the experimental setting is that during the virion treatment the milk concentration is 10-fold higher than in the cell treatment setting. If the milk acts on the virus particles it should therefore be more potent in the virion treatment setting - which was the case in our results. We agree with your prediction, that the milk would not affect already infected cells.
Point 4
Figure 2B – This is a nice experiment. I would suggest repeating with similar milk levels you use in part A. I am very surprised the milk was more effective than SDS and Triton. Showing dose dependence would greatly improve the conclusions one can draw from this experiment.
Response 4: This experimental setup only served as a proof of principle that milk is destroying viral particles. We here used a very high titer of virus to achieve high quantities of measurable RNA, and similarly used high milk concentrations as we did not need to watch out for cytotoxic effects in this setting. Dose dependency study would indeed be interesting; however, we feel that the principle of virus destruction is clear. Whether increasing concentrations of SDS and Triton would result in similar effects as 90% milk, is beyond the scope of this study. We now changed the Figure to better illustrate the degree of destruction.
Point 5
Figure 5 – Can you increase the size of the symbols in the legend. On my print out it was very difficult to determine which lines corresponded to which incubation time.
Expressed breast milk is not sterile and incubating at 37 degrees (and potentially 22 degrees) for 10 hours will cause bacterial growth. How can you conclude the antiviral activity is in the milk or a bi-product of bacterial growth/metabolites? Does the same thing happen when broad spectrum antibiotics are added to the milk to prevent bacterial growth?
Response 5: We now increased the size of the symbols. Additionally, as suggested by reviewer 5, we also depicted the figure as a kinetic over time (Fig S2) and individually as bar diagrams (Fig S3-S6). We have antibiotics present during cell treatment with milk, but not during the incubation of milk. Therefore, we cannot entirely exclude that there is bacterial contamination, however bacterial growth and metabolites at -20°C is unlikely and here we also observed generation of the factor. We discussed this in lines 362-364.
Point 6
How long was the milk frozen in Fig 1? Why does this milk not behave similarly?
Response 6: As commented on in response 1, the milk in Fig 1 was frozen for longer than 6 months. This might explain the milk’s high antiviral potency in comparison to milk that was incubated for maximum of 10 hours. We commented on this in 355-359.
Point 7
In Fig 3 you show the cream portion contains most of the antiviral activity. Did the donor samples have similar fat content?
Response 7: In Fig 3 we analysed the cream portion of stored pooled milk. We did not test individual samples. This experiment served as a proof of principle to test if similar to Pfaender et al., the cream fraction contains the antiviral activity. Milk composition varies in between donors but also within single donors (see references 50+60). Thus, it is likely that the individual differences in milk compositions account for the observed variations (Fig 5). We have now commented on this in lines 382-384.
Point 8
Fig 6 – Here you show virus is killed when kept in the milk at 4 degrees, but not in the freezer. Please discuss why the frozen milk does not do anything now?
Response 8: This is an interesting observation. First, this result suggests that the factor is generated and antivirally active at 4°C. At 4°C lipases show high activity which would allow the generation and activity of the factors.
At -20°C lipases might be less active, but the integrity of the lipid containing micelles is damaged, resulting in lipases getting in contact to a large amount of lipids upon thawing. In Fig 2c we see that thawed milk inactivates ZIKV within 10 minutes. This was the time frame ZIKV was in contact with milk in the Figs 1-5. In the other setup (Fig 6), we thawed the milk containing the high titered virus and immediately titrated it for TCID50 inoculation resulting in only 0.09% milk after thawing. Thus, high virus titers, low milk concentration, and short incubation time at temperatures higher than -20°C might account for the observed differences. We now discussed this in lines 406-416.
Point 9
Line 17 – not sure it is fair to claim Zika causes fatal complications in adults
Response 9: We have now corrected it to “serious”.
Point 10
Line 153 – change to Qiagen
Response 10: Done.
Reviewer 4 Report
The manuscript by Conzelmann et al present data showing that the cream fraction of frozen breastmilk contains anti-ZIKV activity, though fresh breastmilk does not. The data is very clearly presented, and the use of multiple donors enhances the significance of the results. The temperature- and fraction-dependent nature of the antiviral activity is compelling. My only recommendation for improving the manuscript is to expand the discussion regarding the potential factor and how to make sense of some surprising data – the antiviral activity is greatest after the milk has been frozen or refrigerated and incubated for several hours with ZIKV. What is happening to the milk during this time that does not happen at higher temperatures? Would this affect lipase activity?
Author Response
Response to Reviewer 4 Comments
The manuscript by Conzelmann et al present data showing that the cream fraction of frozen breastmilk contains anti-ZIKV activity, though fresh breastmilk does not. The data is very clearly presented, and the use of multiple donors enhances the significance of the results. The temperature- and fraction-dependent nature of the antiviral activity is compelling.
My only recommendation for improving the manuscript is to expand the discussion regarding the potential factor and how to make sense of some surprising data– the antiviral activity is greatest after the milk has been frozen or refrigerated and incubated for several hours with ZIKV. What is happening to the milk during this time that does not happen at higher temperatures? Would this affect lipase activity?
Response 1: We have rewritten large parts of the discussion to address this concern. Specifically, we discussed how the lipoprotein lipase is activated at 4°C and how freezing might affect stability of lipid-protecting micelles and activated bile salt-stimulated lipase. See the full discussion in lines 360-383 and 398-416.
Reviewer 5 Report
The authors investigated the in vitro effect of milk fractions on ZIKV infection on cell culture.
Although the current knowledge shows a low risk for ZIKV postnatal transmission via breast feeding, this is an interesting informative study, which may support public health decisions especially in endemic countries. However, I have some comments and requests.
1) A recent study (Regla-Nava JA et al. PLoS Negl Trop Dis. 2019 Feb 11;13(2):e0007080. doi:10.1371/journal.pntd.0007080) published data on mouse model, which are in line with your research. It would be worth citing it in the introduction or in the discussion.
2) Please indicate if donors signed an informed consent.
3) Are the mothers negative for ZIKV or other flaviviruses serology? Any previous risk exposure or epidemiological link?
4) Are the differences you observed statistically significant? Please include a paragraph with statistics in the Methods and describe also the calculation of IC50s mentioned in the results.
5) Why do you use different concentrations of milk in the experiments? How do you choose the concentrations, for example in Fig 2b (90%) and in 2c (5%)?
6) The effect of the milk on the infection is mostly expressed as E protein detection, except for last experiment; have you measured the infectivity (TICD50) of the treated viral stocks after the exposure to the milk also in the other experiments? The last experiments (considering infectivity) seem to be in contrast with the previous observations at -20°C: in 3.1, 3.2 and 3.3 the milk was stored at -20°C, is it correct? The different observation depends only on the duration of storage? Please discuss better this point.
7) Fig 5 should show kinetics on different concentrations, therefore you should modify the graphs with time on x-axis.
8) Legend of Fig 5c, black line: Should it be 0 and not 0.1?!
9) The line graphs are used to visualize the investigated values over time, generally for experiments of kinetics (such as Fig 5, see above). Therefore, please use bar-graphs to show the effect on the virus on y-axis and milk concentrations on x-axis.
10)I would change the title of captions, with a description of the figure and not of the results.
11)As few cases of transmission by breastfeeding have been reported (i.e. Blohm GM et al., Clin Infect Dis. 2018; Siqueira Mello A et al., Am J Case Rep. 2019), you should be more cautious in the conclusions (lines 366-370).
Author Response
Response to Reviewer 5 Comments
Although the current knowledge shows a low risk for ZIKV postnatal transmission via breast feeding, this is an interesting informative study, which may support public health decisions especially in endemic countries. However, I have some comments and requests.
1) A recent study (Regla-Nava JA et al. PLoS Negl Trop Dis. 2019 Feb 11;13(2):e0007080. doi:10.1371/journal.pntd.0007080) published data on mouse model, which are in line with your research. It would be worth citing it in the introduction or in the discussion.
Response 1: Thank you for this suggestion. We now included it in the discussion. See lines 429-431.
2) Please indicate if donors signed an informed consent.
Response 2: We now indicate in the materials and methods section (lines 87-88) that donors gave informed consent.
3) Are the mothers negative for ZIKV or other flaviviruses serology? Any previous risk exposure or epidemiological link?
Response 3: None of the mothers travelled to regions where ZIKV is endemic. Although we cannot completely rule out exposure to other flaviviruses, it is highly unlikely that antibodies are involved in the observed effects, because they would be antivirally active in fresh milk and would not destroy the viral particle. See lines 88-89 + 360-362.
4) Are the differences you observed statistically significant? Please include a paragraph with statistics in the Methods and describe also the calculation of IC50s mentioned in the results.
Response 4: We added the paragraph explaining how IC50s were calculated (lines 168-171). We decided to not include statistics, as we observed clear dose-dependent inhibition of virus infection and measure infection rates that decrease from 100% to 0% (Figs 1-3, 5) or virus titers from 108 TCID50/ml to undetectable (Fig 6).
5) Why do you use different concentrations of milk in the experiments? How do you choose the concentrations, for example in Fig 2b (90%) and in 2c (5%)?
Response 5: In cell culture experiments, we kept final milk concentrations low (with a maximum of 5%) to minimize or exclude cytotoxic effects. In the experiment shown in Fig. 2b, we analysed the effect of milk in a cell-free system, allowing to increase the concentration. This experimental setup only served as a proof of principle that milk is destroying viral particles
6) The effect of the milk on the infection is mostly expressed as E protein detection, except for last experiment; have you measured the infectivity (TICD50) of the treated viral stocks after the exposure to the milk also in the other experiments? The last experiments (considering infectivity) seem to be in contrast with the previous observations at -20°C: in 3.1, 3.2 and 3.3 the milk was stored at -20°C, is it correct? The different observation depends only on the duration of storage? Please discuss better this point.
Response 6: In all infection experiments, we did not determine the titer of progeny virus. There are two differences in the experimental setups between 3.1-3.3 and Fig 6. First, in 3.1-3.3. we worked with milk that was freeze-stored for >6 months compared to milk that was stored up to 10 days in Fig 6. Second, in the infection setup, we incubated a low-titered virus with milk for 10 minutes before inoculating the cells and determining infection rates. As shown in Fig 2c, within 10 minutes a large fraction of ZIKV is inactivated. In Fig 6 however, we mixed high-titered virus with milk and immediately froze it. Right after thawing the samples were immediately titrated to determine infectivity, which results in a maximum concentration of the milk of 0.09%. Time and concentration might not have been sufficient to inactivated the high ZIKV titers. Thus, high virus titers, low milk concentration, and short incubation time at temperatures higher than -20°C might account for the observed differences. We now discussed this in lines 406-416.
7) Fig 5 should show kinetics on different concentrations, therefore you should modify the graphs with time on x-axis.
Response 7: We now modified the graph and added it as new Figure in the supplement (Fig S2).
8) Legend of Fig 5c, black line: Should it be 0 and not 0.1?!
Response 8: Thank you for pointing out this error. It has now been corrected.
9) The line graphs are used to visualize the investigated values over time, generally for experiments of kinetics (such as Fig 5, see above). Therefore, please use bar-graphs to show the effect on the virus on y-axis and milk concentrations on x-axis.
Response 9: Done (see new Figures S3-S5).
10)I would change the title of captions, with a description of the figure and not of the results.
Response 10: We have now adapted the captions accordingly.
11)As few cases of transmission by breastfeeding have been reported (i.e. Blohm GM et al., Clin Infect Dis. 2018; Siqueira Mello A et al., Am J Case Rep. 2019), you should be more cautious in the conclusions (lines 366-370).
Response 11: We now made clearer, that three cases of transmission have been reported, but that these are rare events which could be explained by an antiviral activity of milk (lines 427-432).
Round 2
Reviewer 3 Report
My concerns have been addressed.